# Insights into Brochosome Distribution, Synthesis, and Novel Rapid-Release Mechanism in *Maiestas dorsalis* (Hemiptera: Cicadellidae)

**DOI:** 10.3390/insects14090734

**Published:** 2023-08-30

**Authors:** Wei Wu, Jia-Ning Lei, Qianzhuo Mao, Yan-Zhen Tian, Hong-Wei Shan, Jian-Ping Chen

**Affiliations:** State Key Laboratory for Managing Biotic and Chemical Threats to the Quality and Safety of Agro-Products, Key Laboratory of Biotechnology in Plant Protection of Ministry of Agriculture and Zhejiang Province, Institute of Plant Virology, Ningbo University, Ningbo 315211, China

**Keywords:** leafhopper, brochosome, *Maiestas dorsalis*, TEM, SEM

## Abstract

**Simple Summary:**

Leafhoppers, a large hemipteran family, have a unique behavior of covering themselves with brochosomes. Brochosomes are believed to act as a protective coating for leafhoppers, serving to enhance hydrophobicity and provide protection against predators and parasites. In this study, we investigated the distribution, synthesis, and release mechanisms of brochosomes in the rice pest, *Maiestas dorsalis*. We found uniform brochosome coverage on different body parts and observed the step-by-step process of brochosome synthesis in the Malpighian tubules. Additionally, we identified a rapid and efficient mechanism for the release of brochosomes. Our study provides valuable insights into the synthesis and release mechanisms of brochosomes, enhancing our understanding of leafhopper biology.

**Abstract:**

The leafhopper family Cicadellidae, comprising over 22,000 species, exhibits a unique behavior of anointing their bodies with excretions containing brochosomes. Brochosomes are synthesized in the distal segment of the Malpighian tubules and serve various functions, including hydrophobic protection and defense against pathogens and predators. In this study, we investigated the distribution, synthesis, and release mechanisms of brochosomes in the rice pest leafhopper *Maiestas dorsalis*. Using SEM and TEM, we observed brochosomes’ consistent coverage on the integument throughout the insect’s life cycle. Moreover, we identified four distinct developmental stages of brochosome synthesis within the distal segment of the Malpighian tubules, originating from the Golgi region. Most importantly, our research revealed a novel and highly efficient release mechanism involving the fusion of brochosome-containing vesicles, leading to a rapid and substantial release of brochosomes into the tubule lumen after molting. These findings shed light on the intricate processes of brochosome synthesis and release in leafhoppers, offering valuable insights into their functional significance and ecological role in these fascinating insects.

## 1. Introduction

The leafhopper family Cicadellidae, which comprises over 22,000 described species, is one of the largest families of Hemiptera [1]. Leafhoppers exhibit a unique behavior known as anointing, where they coat their bodies with excretions [2,3,4]. Brochosomes are the main component of these excretions, and are commonly present on the integument of almost all leafhoppers [5,6]. Brochosomes have also been found in the integument of insects from the Hemiptera, Diptera, and Hymenoptera orders; most of these occurrences have been confirmed as resulting from contamination rather than being produced by the insects themselves [7,8,9]. The functions of brochosomes have been presumed to include hydrophobic protection against dew and excretions, and defense against pathogens, predators, parasites, and other environmental factors [5,6]. However, while they have been shown to provide hydrophobicity and protection against predators and parasites, other proposed functions lack biological experimental support [10,11,12,13].

Brochosomes are synthesized in specific glandular segments of the Malpighian tubules in leafhoppers [5,6]. In insects, the Malpighian tubules serve as the main excretory organ responsible for the removal of metabolic waste and the regulation of water and ion balance [14]. Within Cicadomorpha, distinct segments of the Malpighian tubules have undergone evolutionary adaptations that confer secretory functions [15,16,17]. The Malpighian tubules in leafhoppers can generally be divided into three segments based on their morphology: proximal, distal, and terminal. The proximal and distal segment are elongated, with epithelial cells featuring a brush-like inner border rich in slender microvilli, and cytoplasm containing a widespread endoplasmic reticulum and numerous mitochondria. The distal segment, characterized by its rod-like and swollen structure, is responsible for brochosome synthesis [18,19,20]. It is lined by epithelial cells that exhibit typical secretory characteristics, including large spherical nuclei, extended rough endoplasmic reticulum, and numerous Golgi vesicles [18,21,22]. The cytoplasm of these epithelial cells contains numerous vesicles containing developing or fully matured brochosomes [18,21,22]. Brochosomes originate in the Golgi region and continue to develop within Golgi vesicles, with increasing granule diameters and progressively deeper surface invaginations, eventually developing into mature brochosomes [21,22]. Mature brochosomes may exhibit variations in morphology among different leafhopper species, but they typically appear as hollow microspheres with a diameter of 0.2–0.7 μm, featuring a honeycomb-like surface resembling footballs or buckminsterfullerene (C60) molecules [5,6]. After maturation, the brochosomes are released into the Malpighian tubule lumen and eventually excreted through the hindgut [5,6,17,18,19].

After each molt, leafhoppers apply brochosomes as a coating onto their fresh integument (anointing). This anointing behavior typically occurs 1–3 h after molting [3]. Leafhoppers use their hind legs to spread the secretion containing brochosomes onto their integument. After the liquid component dries, they continuously groom their body using their hind legs to evenly distribute the brochosomes [3,20]. Moreover, due to the continuous shedding of brochosomes during the grooming process and in daily activities, this anointing behavior might be repeated multiple times between two molts to replenish the shed brochosomes on the body surface [6,20]. Previous research has suggested that brochosomes merge with the apical plasma membrane through brochosome-containing vacuoles and are released into the lumen, following a process similar to exocytosis [18,21,23,24]. Additionally, the leafhopper’s Malpighian tubules lack a dedicated storage structure for brochosomes. The mature brochosomes reside within epithelial cells and are released into the Malpighian tubule lumen only before molting [3]. However, the reliance on the “exocytosis” mechanism to release brochosomes is limited to releasing only those brochosomes contained within the vacuoles near the plasma membrane edge. This constraint may not adequately meet the leafhoppers’ requirement for rapid and substantial brochosome release after each molt, especially considering their significant need for brochosomes to coat the new exoskeleton post-molting. Given this context, utilizing “exocytosis” for releasing brochosomes might be more appropriate for replenishing the lost brochosomes on the body surface after the initial anointing behavior following molting. Thus, apart from the exocytosis-like mechanism, there is a possibility of alternative mechanisms that could facilitate swift and abundant brochosome release.

*Maiestas dorsalis* (Motschulsky, 1859), a common pest widely distributed in Asian rice-growing regions, directly feeds on rice plants and acts as a vector for various rice pathogens [25,26]. Similar to other insects in the Cicadellidae family, *M. dorsalis* possesses a brochosome-coating integument. However, research on the distribution, synthesis, and release mechanisms of brochosomes in *M. dorsalis* has not been conducted. In this study, we used light microscopy to observe the composition of *M. dorsalis*’ digestive system, including the foregut, anterior diverticulum, midgut, and Malpighian tubules (Figure 1A,B). The Malpighian tubules consist of slender proximal and terminal segments and a swollen rod-like distal segment (Figure 1C). We also examined the distribution of brochosomes on different body parts during nymph and adult developmental stages. Additionally, we employed transmission electron microscopy to observe the synthesis and secretion process of brochosomes in the epithelial cells of the distal segment of the Malpighian tubules. By comparing samples at different time points after molting, we identified an efficient mechanism for the massive release of brochosomes. The results of this study will contribute to a deeper understanding of brochosome synthesis and release mechanisms in leafhoppers.

## 2. Materials and Methods

### 2.1. Insect Rearing

Adult leafhopper *M. dorsalis* were collected from a rice field in Jiaxing, Zhejiang Province, China, in September 2020. Subsequently, the leafhoppers were reared in an insect-proof greenhouse for over two years at 26 ± 1 °C, with a 16:8 h light-to-dark cycle and 50 ± 5% relative humidity. TaiChung Native 1 (TN1) rice was grown under the same conditions for leafhopper feeding.

### 2.2. Light Microscopy

Live individuals of *M. dorsalis* were frozen-anesthetized on ice for 10 min, and the alimentary tract and Malpighian tubules were dissected out in phosphate-buffered saline solution (PBS, 0.1 M, pH = 7.2), under a Stereoscopic Zoom Microscope ZSA0850 (COIC, Chongqing, China). Photos were imaged by using a Nikon SMZ225 stereo microscope (Nikon, Tokyo, Japan) and a Ds-Ri2 digital camera (Nikon, Tokyo, Japan).

### 2.3. Scanning Electron Microscopy

In order to investigate the brochosomes present on the integument of leafhopper *M. dorsalis*, a scanning electron microscope (SEM) was employed. Different developmental stages of *M. dorsalis* nymphs and adults were collected, and the adults’ forewings, hindwings, and hind legs were dissected under a stereomicroscope. The collected samples were subsequently subjected to a drying process in a vacuum oven at 50 °C for 24 h. The dried samples were affixed onto specimen holders using a conductive adhesive and coated with a layer of gold using a sputter coater. Subsequently, the samples were examined and subjected to microscopic imaging using an S-4800 scanning electron microscope (Hitachi, Tokyo, Japan) at an accelerating voltage of 3.0 kV.

### 2.4. Transmission Electron Microscopy

Leafhoppers apply brochosomes onto their newly molted integument after each molt, and they possess the capacity to synthesize brochosomes throughout their entire lifespan. In this study, we aimed to elucidate the synthesis and release mechanisms of brochosomes in *M. dorsalis*. We dissected and collected Malpighian tubule tissues from various nymphal stages and adult leafhoppers at 0 (*M. dorsalis* is shedding its old skin and has not yet coated the new integument with brochosomes) and 6 h (*M. dorsalis* has already coated the new integument with brochosomes) after molting. We then used transmission electron microscopy to observe the process of brochosome synthesis and secretion in the leafhopper *M. dorsalis.*

The samples were fixed initially with a solution of 2% paraformaldehyde and 2.5% glutaraldehyde in PBS buffer at 4 °C overnight. After three washes with PBS buffer, tissues were further fixed with 2% osmium tetroxide in PBS buffer at 4 °C overnight. Following three additional washes in PBS buffer, the fixed tissues underwent dehydration using a graded series of ethanol concentrations (30%, 50%, 70%, 80%, 90%, 95%, and 100%) for 20 min at each step. Subsequently, the samples were transferred to absolute acetone for an additional 20 min. The next step involved immersing the samples in a 1:1 mixture of Spurr resin and absolute acetone at room temperature for 1 h, followed by transfer to a 3:1 mixture of Spurr resin and absolute acetone at room temperature for 3 h. Finally, the samples were placed in a mixture of Spurr resin overnight. The prepared samples were embedded in capsules containing an embedding medium and heated at 70 °C overnight. For visualization of the specimen sections, they were stained with uranyl acetate and alkaline lead citrate for 5–10 min each. The resulting images were observed using a Hitachi electron microscope HT7800 (Hitachi, Tokyo, Japan).

## 3. Results

### 3.1. Brochosomal Coatings on the Integument of Leafhopper M. dorsalis

The brochosomes on the integument of *M. dorsalis* exhibit the typical morphology found in most leafhopper species. They are spherical with a diameter of approximately 450 nm and possess a honeycomb-like surface (Figure 2B,C). The surface structure is composed of pentagonal and hexagonal indentations, each with a central pore (Figure 2B,C). Observation of brochosome distribution on the integument, forewing, hindwing, middle leg, and hind leg of *M. dorsalis* adults revealed uniform coverage across all body parts, with no morphological differences in brochosomes among distributions (Figure 2A,D–H). Furthermore, examination of brochosome distribution at different developmental stages of *M. dorsalis* demonstrated brochosome coverage on both nymphs and adults, but no brochosome distribution was observed on the surface of the eggs of *M. dorsalis* (Appendix A).

### 3.2. Brochosome Genesis in the Leafhopper M. dorsalis

To observe the synthesis process of brochosomes in the leafhopper *M. dorsalis*, we first examined the structure of the proximal segment, distal segment, and terminal segment of the Malpighian tubule in adult *M. dorsalis* (Figure 3A). The proximal segment and terminal segment of the Malpighian tubule have similar structures (Figure 3B,D). The basement membrane exhibits distinct folds and is densely covered with microvilli near the lumen. The cytoplasm of epithelial cells is filled with a large number of mitochondria (Figure 3B,D). Free brochosomes were observed in the lumen of the terminal segment but not in the proximal segment (Figure 3B,D). In contrast, the distal segment of the Malpighian tubule was uniformly thicker with a smooth basement membrane and no microvilli near the lumen (Figure 3C). The epithelial cells of the distal segment exhibited typical secretory-cell characteristics, with a spherical nucleus containing heterochromatin clumps and surrounded by abundant rough endoplasmic reticulum and Golgi. The cytoplasm contained numerous vesicles housing developing or fully mature brochosomes (Figure 3C and Figure 4A).

The Golgi region serves as the site of origin for brochosomes, with numerous vesicles of varying sizes present in this region. Different developmental stages of brochosomes can be observed within the vesicles near the Golgi area, indicating a temporal sequence of brochosome formation (Figure 4B–E). In the initial state, each brochosome is individually enclosed within a vesicle, with a diameter of approximately 300 nm and an irregular shape with shallow depressions on the surface (Figure 4F). These brochosomes undergo continuous development, with an increasing diameter and deepening surface depressions. These depressions gradually deepen, forming regular invaginations, while the central region of the brochosome exhibits high electron density, appearing black (Figure 4G). As the brochosomes continue to develop, the invaginations on their surface deepen, reaching approximately one-fourth of the particle diameter. At this stage, the brochosomes reach their final size, and both the particle diameter and the number of surface depressions do not increase during late development (Figure 4H). At this stage, the brochosomes exhibit a loosely structured flocculent core and a dense wall (Figure 4C,H). Subsequently, multiple Golgi vesicles merge to form larger vesicles containing several developing brochosomes (Figure 4D). The outer edges of the brochosome septa expand, the flocculent material in the central region disappears, and a transparent matrix replaces it (Figure 4I). Within the brochosome-containing vesicles (BVs), the brochosomes reach complete maturity, with the dense material in the central cavity disappearing, and a hole appears at the bottom of the invagination, extending directly to the center (Figure 4E,J). Moreover, the structure of Malpighian tubules in *M. dorsalis* is entirely consistent across all segments in both the nymphal and adult stages, and the process of brochosome synthesis at the distal segment of the Malpighian tubule is also identical.

### 3.3. Brochosome Release from Epithelial Cells into the Lumen of the Malpighian Tubule

By continuously observing samples of *M. dorsalis* at various times after molting, including 2nd to 5th instar nymphs and adults, we systematically investigated the process of brochosome release from epithelial cells to the lumen of the distal Malpighian tubule segments. Initially, brochosomes were released into the Malpighian tubule lumen through fusion between brochosome-containing vesicles (BVs) and the apical plasma membrane proximal to the lumen (Figure 5A). This phenomenon was observed in almost all freshly molted samples and in a small number of leafhopper samples that had molted for more than a day. Furthermore, in freshly molted leafhopper samples, we observed a more efficient mode of brochosome release. In adult samples collected at 0 h post-molt, continuous fusion among BVs in the vicinity of the initial release site resulted in the continuous release of brochosomes into the tubule lumen (Figure 5B,C). As BVs continually fused, the majority of BVs containing mature brochosomes within the epithelial cells merged, forming a large cavity that released a substantial quantity of brochosomes into the tubule lumen (Figure 5D). Consequently, a significant distribution of free-floating brochosomes was observed in the Malpighian tubule lumen (Figure 5D,E). Ultimately, in adult samples collected at 6 h post-molt, we observed that only a small quantity of brochosomes remained within the cavities formed by the fusion of BVs (Figure 5F), indicating the completion of brochosome release. Additionally, this cavity formed by the fusion of multiple BVs was observed not only in freshly molted adult samples but also in freshly molted nymph samples (Appendix A). These findings indicate that leafhoppers can rapidly release stored brochosomes into the tubule lumen after each molt through BV fusion, ensuring an ample supply of brochosomes for integument application.

## 4. Discussion

Leafhoppers exhibit a unique behavior known as anointing, wherein after each molt, they actively apply protein–lipid particles known as brochosomes onto their integument [2,3,4,5,6]. Brochosomes are considered as a protective coating on the leafhopper’s integument, proven to enhance hydrophobicity and play a significant role in resisting predators and parasites [10,11,12,13]. In this study, we delineate the distribution of brochosomes on the body surface of *M. dorsalis* and the process of synthesis and release. Through a continuous observation of samples at various time points post-molt, we discovered a rapid and efficient brochosome release mechanism, which might be linked to the leafhopper anointing behavior.

Brochosomes form a unique covering found on the integument of Cicadellidae. This covering has been identified across various subfamilies and major tribes within the Cicadellidae family [5,6]. Most leafhopper species produce and apply brochosomes to their integument during both nymph and adult stages [5,6]. In certain leafhopper species, brochosome production commences during the last nymphal instar, leading to the release and deposition of brochosomes onto the integument only after the final molt [19]. Across studied leafhopper species, brochosome morphology consists of hollow spheres. Their size and arrangement exhibit variation among species, developmental stages, genders, and individual particles within the same leafhopper [5,6]. Nevertheless, in most examined species, intra-brochosome structures share a high degree of similarity, measuring 0.2–0.7 μm in diameter, and displaying a surface reminiscent of honeycombs, akin to footballs or buckminsterfullerene (C60) molecules [5,6]. In *M. dorsalis*, the brochosomes were observed to be spherical, approximately 450 nm in diameter, and characterized by a honeycomb-like surface texture (Figure 2B,C). Across the complete life cycle of *M. dorsalis*, the integument consistently maintains a layer of brochosomes, thereby preserving a uniform morphology, which concurs with the prevailing literature on brochosome behavior in leafhoppers (Figure 2D–H and Appendix A). Some female adults of the tribes Proconiini and Phereurhinini (Cicadellidae: Cicadellinae) utilize brochosomes to powder their eggs and oviposition sites, and the egg brochosomes have morphological distinctions from the common integumental brochosomes [27]. However, we did not observe the presence of brochosomes either at the egg-laying site or on the surface of eggs in *M. dorsalis*.

Brochosomes are generated within specialized glandular segments of the Malpighian tubules. In most insects, the Malpighian tubules serve as the main excretory organ, responsible for excreting metabolic waste and regulating ion concentrations and water balance [14,28]. In Cicadomorpha, specific regions of the Malpighian tubules have evolved secretory functions [15,16,17]. For instance, Cicadoidea insects utilize Malpighian tubule secretions to reinforce their burrows, and Cercopoidea insects produce a foamy substance for protection [15,16,17,29]. In Cicadellidae, the Malpighian tubules secrete brochosomes, which form an outer coating on their integument [5,6]. Unlike the secretory products in Cicadoidea and Cercopoidea, the brochosomes synthesized and secreted by the Malpighian tubules of leafhoppers exhibit specific morphologies and complex internal structures, suggesting a more intricate processing mechanism. The distal segment of the leafhopper’s Malpighian tubule exhibits specific enlargement, surrounded by a layer of epithelial cells, each of which synthesizes a large number of brochosomes [18,21,22]. In leafhopper *M. dorsalis*, brochosomes originate from the Golgi region of the epithelial cells (Figure 4A,B). We observed five distinct developmental stages of brochosomes in the Golgi vesicles and brochosome-containing vesicles (Figure 4C,D). In the initial stage, the brochosome is a solid particle, with an irregular shape and a diameter of approximately 300 nm, and its surface shows shallow depressions. As the brochosome continues to develop, its diameter increases, and the surface depressions deepen. Ultimately, the mature brochosomes exhibit a completely clear central cavity and a central invagination (Figure 4F–J). In leafhopper *Oncometopia orbona* (Fabricius, 1798), it was shown that brochosomes originate from Golgi saccules with a diameter of about 0.06 µm [21]. However, in our study, we did not observe this initial form of brochosomes.

After each molt, leafhoppers apply a layer of brochosomes onto their new integument. Notably, the Malpighian tubules of leafhoppers lack any structures for storing brochosomes, suggesting that these are stored inside cells until they are quickly discharged into the lumen and pumped through the proximal parts of the Malpighian tubules to the hindgut and rectum shortly before the leafhopper extrudes droplets of secretory fluid and applies them onto the body surface [3]. In leafhoppers such as *Psammotettix striatus* (Linnaeus 1758) and *Graphocephala fennahi* (Young, 1977), studies have demonstrated that the brochosomes released into the Malpighian tubule lumen in a cytosolic-like manner through BVs merge with the apical plasma membrane [18,21,23,24]. However, this “exocytosis” process seems applicable only to brochosomes housed within vacuoles situated near the plasma membrane periphery. Consequently, this method might not satisfactorily cater to the leafhoppers’ requirement for swift and substantial brochosome release immediately after molting, particularly considering the necessity to coat the new exoskeleton. In this study, we have uncovered a novel and highly efficient mechanism for the rapid release of brochosomes. In newly molted *M. dorsalis*, brochosomes are initially discharged into the Malpighian tubule lumen through the fusion between brochosome-containing vesicles (BVs) and the apical plasma membrane in proximity to the lumen (Figure 5A). This initial release sets the stage for the subsequent coalescence of BVs around the release site, resulting in the formation of a spacious cavity. Ultimately, nearly all BVs containing mature vesicles within individual epithelial cells merge, culminating in the creation of a substantial cavity (Figure 5B,C). This process of multiple BVs merging to create a large cavity is a recurring phenomenon observed in Malpighian tubule samples of *M. dorsalis* across various developmental stages shortly after molting (Figure 5D–F and Appendix A). This phenomenon could have been overlooked in prior studies due to the possibility that the mechanism involving substantial brochosome release through BV fusion might be unique to recently molted leafhoppers. Previous research might not have concentrated on examining leafhoppers shortly after molting, potentially causing the omission of this specific mechanism.

Additionally, an intriguing observation emerged when examining certain *M. dorsalis* Malpighian tubule samples that had molted for over a day. In these cases, we observed a small-scale release of brochosomes into the Malpighian tubule lumen through a mechanism resembling cytosolic exocytosis. We propose that this limited brochosome release could be aimed at compensating for the loss of brochosomes from the body surface. After anointing, leafhoppers use their hind legs for continuous grooming to ensure uniform brochosome coverage on the body surface [3]. During this grooming process, brochosome detachment might occur. This phenomenon is further supported by the fact that some leafhoppers store brochosomes along the edges of their wings [3,4,30]. Additionally, daily activities could lead to brochosome shedding, such as when leafhoppers come into contact with liquids on their body surface or become entangled in spider webs [10,11,12]. Furthermore, the presence of brochosomes has been detected in the atmosphere [31,32]. The process of brochosome shedding during grooming and daily activities could result in localized gaps in brochosome coverage on the leafhopper’s body surface. As a result, it is conceivable that multiple anointing behaviors occur between two consecutive molts to replenish the lost brochosomes. Based on these observations, we propose the existence of two distinct brochosome release mechanisms in leafhoppers. The first involves the rapid and substantial release of brochosomes through BV fusion, meeting the initial anointing requirements following molting. The second mechanism employs a process similar to exocytosis, releasing a smaller amount of brochosomes located near the tubular plasma membrane. This mechanism serves to supplement the loss of brochosomes from the body surface during subsequent anointing behaviors.

In conclusion, our study provides valuable insights into the distribution, synthesis, and release mechanisms of brochosomes in the rice pest leafhopper, *M. dorsalis*. We observed uniform brochosome coverage on different body parts of the leafhopper throughout its life cycle, confirming the protective coating role of brochosomes. Through detailed examination of the distal segment of the Malpighian tubules, we elucidated the step-by-step process of brochosome synthesis, originating from the Golgi region and developing within vesicles with increasing complexity. Notably, we discovered a novel and efficient release mechanism involving the fusion of brochosome-containing vesicles, resulting in rapid and substantial release of brochosomes into the tubule lumen immediately after molting. This mechanism ensures an ample supply of brochosomes for integument application, particularly for the initial anointing behavior following molting. Our findings contribute to a deeper understanding of leafhopper biology and their unique behaviors, shedding light on the intricate processes of brochosome synthesis, distribution, and release. Further research in this area could explore the functional significance of different brochosome release mechanisms and investigate their adaptive value in different ecological contexts.

## Figures and Tables

**Figure 1 insects-14-00734-f001:**
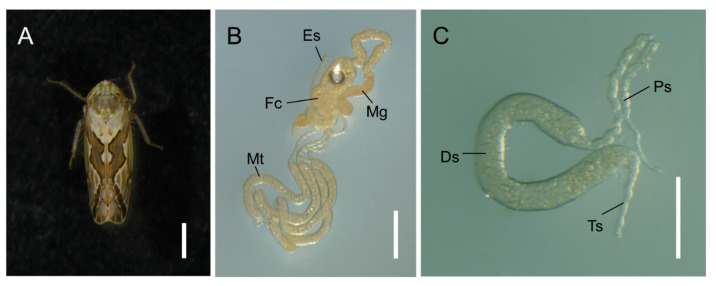
Adult male leafhopper *M. dorsalis* and its alimentary tract and Malpighian tubules. (**A**) A male of *M. dorsalis*. Bar, 1 mm. (**B**) Light micrograph of the alimentary canal in *M. dorsalis*. Bar, 1 mm. (**C**) Structure of the Malpighian tubule. Bar, 1 mm. Es, esophagus; Fc, filter chamber; Mg, midgut; Mt, Malpighian tubules; Ds, distal segment; Ps, proximal segment; Ts, terminal segment. All images are representative of at least three replicates.

**Figure 2 insects-14-00734-f002:**
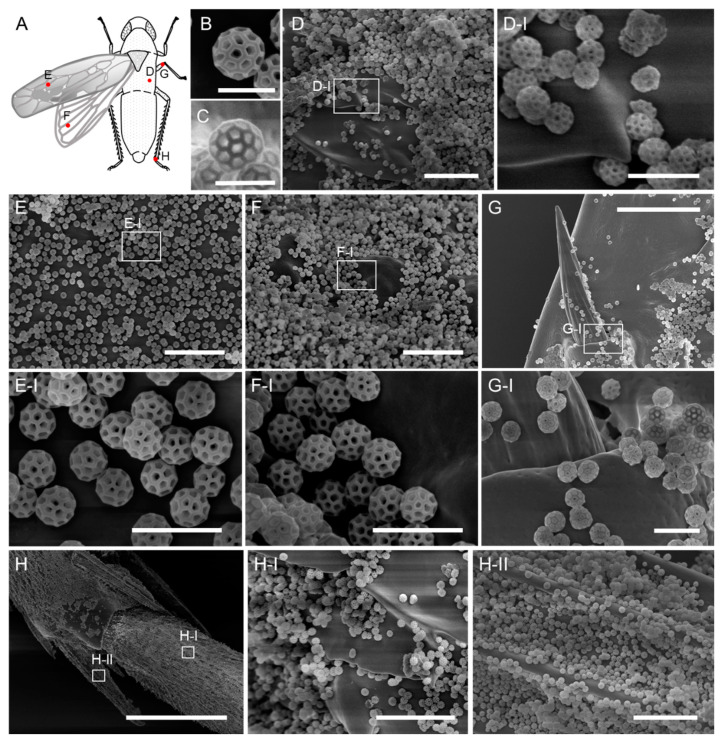
Brochosomal coatings on the integument of adult leafhopper *M. dorsalis*. (**A**) Cartoon picture of adult leafhopper *M. dorsalis*. Right wings are not shown. Red dots indicate scanning electron microscope observation positions of D–H. (**B**,**C**) A single brochosome on the integument of leafhopper *M. dorsalis*. Bar, 500 nm. (**D**–**H**) Brochosome distribution on the integument (**D**,**D-I**), forewing (**E**,**E-I**), hindwing (**F**,**F-I**), middle leg (**G**,**G-I**), and hind leg (**H**,**H-I**,**H-II**) of adult leafhopper *M. dorsalis*. **D-I**,**E-I**,**F-I**,**G-I**,**H-I**,**H-II** are enlargements of the boxed regions in **D**,**E**,**F**,**G**,**H**, respectively. Scale bars in **D-I**,**E-I**,**F-I**,**G-I**, 1 µm; **D**,**E**,**F**,**H-I**,**H-II**, 5 µm; **G**, 10 µm; **H**, 100 µm.

**Figure 3 insects-14-00734-f003:**
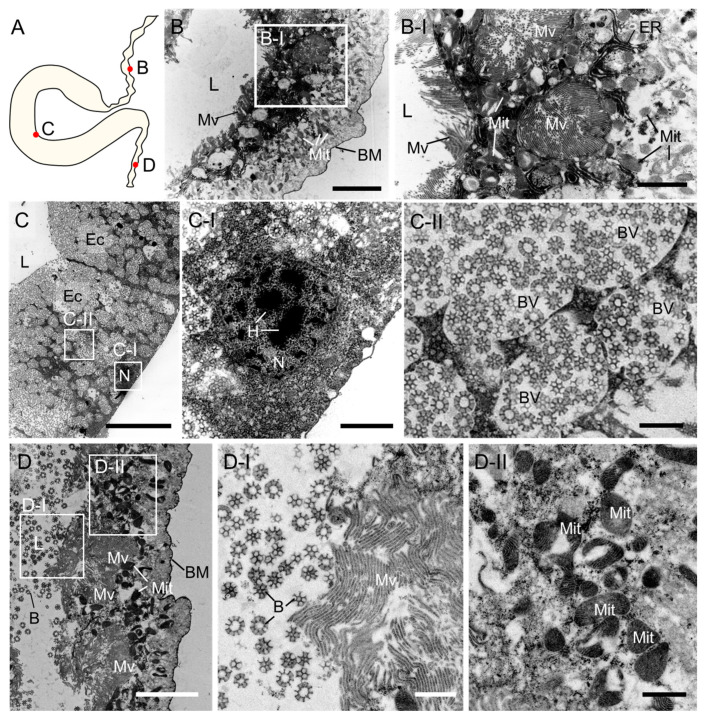
Cellular structure of the proximal, distal, and terminal segments of the Malpighian tubules of adult leafhopper *M. dorsalis*. (**A**) Cartoon picture of the *Malpighian tubules.* (**B**) Cross-section of the proximal Malpighian tubule segments. The proximal segment of the Malpighian tubule is extremely thin, and the basement membrane exhibits distinct folds (**B**). It is densely covered with microvilli near the lumen. The cytoplasm contains abundant mitochondria and a widely distributed endoplasmic reticulum (**B-I**). **B-I** is enlargement of the boxed area in **B**. Scale bars in **B**, 5 µm; **B-I**, 2 µm. (**C**) Cross-section of the distal Malpighian tubule segments. The distal segment of the Malpighian tubule was uniformly thicker with a smooth basement membrane and no microvilli near the lumen (**C**). Epithelial cells with a spherical nucleus containing heterochromatin clumps (**C-I**). Cytoplasm contained numerous vesicles housing developing or fully mature brochosomes (**C-II**). C-I and **C-II** are enlargements of the boxed area in **C**. Scale bars in **C**, 20 µm; **C-I**, 2 µm; **C-II**, 1 µm. (**D**) Longitudinal section of the terminal Malpighian tubule segments. The distal segments of the Malpighian tubule have a similar structure to the proximal segments. The basement membrane exhibits distinct folds and is densely covered with microvilli near the lumen (**D**). A significant amount of free-floating brochosomes can be observed within the tubule lumen (**D-I**). The cytoplasm contains abundant mitochondria (**D-II**). **D-I**,**D-II** are enlargements of the boxed area in **D**. Scale bars in **D**, 5 µm; **D-I**,**D-II**, 1 µm. B, brochosome, BM, basement membrane; BVs, brochosome-containing vesicles; Ec, epithelial cell; ER, endoplasmic reticulum; H, heterochromatin; L, lumen; Mv, microvilli; Mit, mitochondria; N, nucleus. All images are representative of at least three replicates.

**Figure 4 insects-14-00734-f004:**
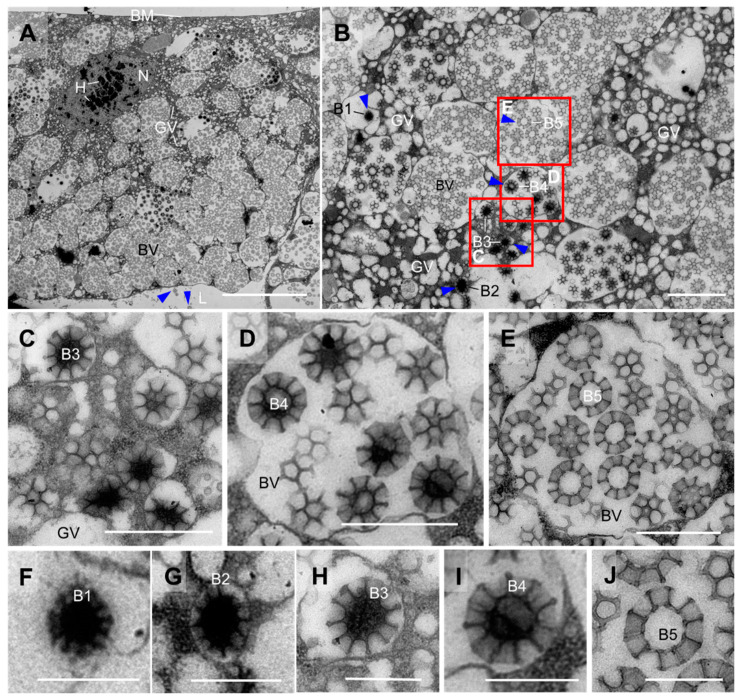
Brochosome synthesis in epithelial cells of distal Malpighian tubule segments of adult leafhopper *M. dorsalis*. (**A**) An epithelial cell located in the distal segment of the Malpighian tubule of leafhopper *M. dorsalis*. The spherical nucleus is characterized by clusters of heterochromatin, which are surrounded by a substantial amount of rough endoplasmic reticulum and Golgi apparatus in close proximity to the nucleus. The cytoplasm is occupied by numerous vesicles containing varying numbers of brochosomes. Additionally, there are some free brochosomes (blue arrow) in the lumen of the Malpighian tubule. Bar, 10 µm. (**B**–**I**) TEM observation of the brochosome synthesis region in the epithelial cells of the distal segment of the Malpighian tubule. Brochosomes are synthesized within Golgi vesicles (**C**–**E**). Initially, brochosomes have an irregular shape (**F**). As they develop, they form surface indentations, and the central matrix gradually becomes filamentous (**G**–**I**). Mature brochosomes have a well-defined outer edge and a central region with a noticeable cavity (**J**). **C**–**E** are enlargements of the boxed area in **B**. Scale bars in **B**, 2 µm; **C**–**E**, 1 µm; **F**–**I**, 500 nm. B1–B3, developing brochosome in Golgi vesicles; B4, developing brochosome in brochosome-containing vesicles; B5, fully mature brochosome; BM, basement membrane; BVs, brochosome-containing vesicles; H, heterochromatin; L, lumen; N, nucleus; GV, Golgi vesicle. All images are representative of at least three replicates.

**Figure 5 insects-14-00734-f005:**
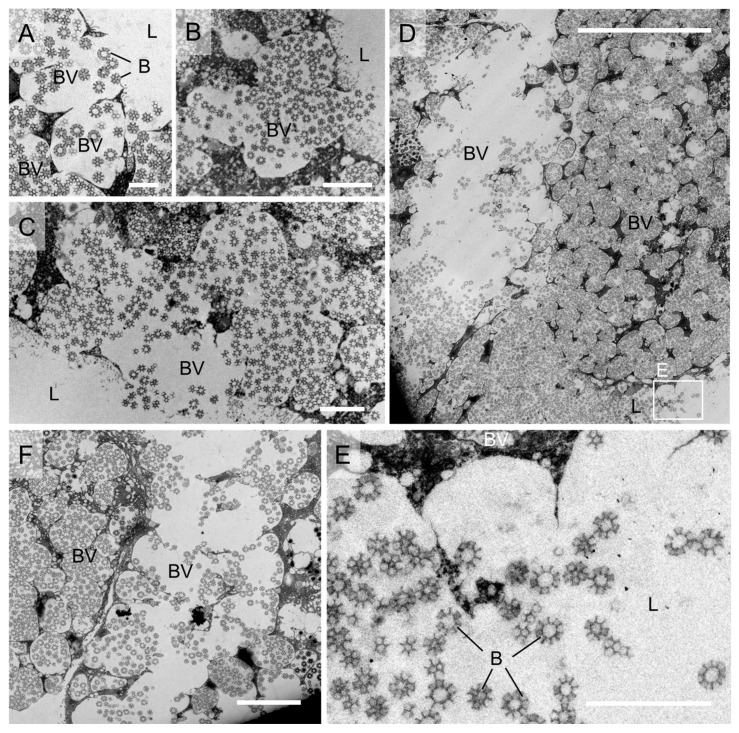
Brochosome release from epithelial cells of distal Malpighian tubule segments of adult leafhopper *M. dorsalis*. (**A**) BVs near the lumen merge with the apical plasma membrane, releasing brochosomes into the tubule lumen. Bar, 1 µm. (**B**,**C**) BVs adjacent to the initial brochosome release site fuse with each other, releasing brochosomes from the neighboring BVs into the lumen. Scale bars in **B**,**C**, 2 µm. (**D**,**E**) BVs in the distal segment epithelial cells fuse, releasing a large quantity of brochosomes into the tubule lumen. E is enlargement of the boxed area in **D**. Scale bars in **D**, 20 µm; **E**, 5 µm. (**F**) Eventually, within a single epithelial cell of the distal segment of the Malpighian tubule, almost all the BVs containing mature brochosomes fuse with each other. Scale bars in **E**, 5 µm. B, brochosome; BVs, brochosome-containing vesicles; L, lumen. All images are representative of at least three replicates.

## Data Availability

The data presented in this study are available in the article and Appendix A.

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
