# Peer review of "Insights into Brochosome Distribution, Synthesis, and Novel Rapid-Release Mechanism in Maiestas dorsalis (Hemiptera: Cicadellidae)"

_insects, 2023, doi:10.3390/insects14090734_

Round 1
Reviewer 1 Report (Previous Reviewer 1)
Authors show the efficient release of brochosome occurs at each molt in this species and also show the distribution of released brochosomes on the body. This MS can be acceptable after very minor corrections.
1. The stage (adult or nymph) are not clearly shown in Figure legend ( Fig.2-5) and text of result. It should be indicated in the legend and text.
2. In L351 (data not shown) or (unpublished data) should be added.
Author Response
Authors show the efficient release of brochosome occurs at each molt in this species and also show the distribution of released brochosomes on the body. This MS can be acceptable after very minor corrections.
Response: We would like to express our gratitude for your careful review of our manuscript. Your insightful comments and suggestions have been incredibly valuable in refining our work. We are pleased to respond to your feedback with the following revisions:
- The stage (adult or nymph) are not clearly shown in Figure legend (Fig.2-5) and text of result. It should be indicated in the legend and text.
Response: Thank you for your comments. In response to your request, we have incorporated the descriptions of sample sources and their corresponding periods in the captions of Figures 2-5. Furthermore, we have included information about the sample sources and their corresponding periods in appropriate sections of the manuscript's results. Your guidance has been invaluable in enhancing the clarity and comprehensiveness of our paper.
- In L351 (data not shown) or (unpublished data) should be added.
Response: Thank you for bringing up the need to address the matter related to the mention of "data not shown." We have taken your suggestion into account and added "data not shown" at line 351. Your input has been instrumental in improving the transparency and completeness of our study.
Reviewer 2 Report (Previous Reviewer 2)
No comments.
Author Response
We greatly appreciate your recognition and evaluation of our manuscript. We are delighted that the reviewer found no comments or suggestions regarding our work. This positive feedback indicates that our research content and presentation have been acknowledged. Such affirmation serves to inspire us to delve deeper into research and exploration. We extend our gratitude for your valuable time and feedback, and we eagerly anticipate sharing this achievement with the academic community.
This manuscript is a resubmission of an earlier submission. The following is a list of the peer review reports and author responses from that submission.
Round 1
Reviewer 1 Report
Authors should re-consider the style of discussion as you never show the figure number in the text. Such a large fusion event (small fusion events were already reported in other papers) may be the only new result in this MS, but you show the result only on adult eclosion. So you should not say that this event occurs at each molt and also discuss why such a fusion event is found only in this species. If this event occurs only in this species, you should describe the physiological meaning of large fusion in this species but not in other species.
1. L117-119, These sentences should be move to introduction.
2. In Fig.3 This is the only new result as compared with previous reports. However, this fusion event also occurs when the fixtave is changed to other ones?
3. In discussion, you should indicate the figure number. We don't know whether the results are reported in previous paper or this paper.
4. If this fusion events are new, you should compare the previous paper. Such a large release of brochome occurs only in this insect? Do you mean that small amount of brochome are released in other insects? If so you should show the large amount of excresiton of brochosomes outside of the body in this species.
5. You describe that such a fusion events occurs at each molt in this species, but you show only the result on adult eclosion (Fig.3). Is there any proof that such a fusion event occurs at each molt? In discussion you used "each molt" too, but this is the only speculation.
6. Fusion events already shown in the previous papers (in Fig.7, Yuan and Wei,BMC Genomics. 2022 Jan 21;23(1):67.; inFig.20,Rakitov et al. Insect Biochemistry and Molecular Biology 2018, 94, 10-17) although they did not describe the fusion event in detail. You should describe fusion events are already suggested in a previous paper although such a dynamic fusion had not been reported.
7. In L288-289, when and in which species this method release occur? There is no citation.
Moderate editing may be required.
Reviewer 2 Report
This paper touches on a significant issue regarding the brochosomes in a Cicadellidae species. Because research on this structure is still unsatisfactory, I suggest publishing the MS after several essential corrections.
Most important is the erroneous taxonomic position of the species used for the study. According to the recent revision of the Deltocephalini (Cicadellidae) [Webb & Viraktamath, 2009: Annotated check-list, generic key and new species of Old World Deltocephalini leafhoppers with nomenclatorial changes in the Deltocephalus group and other Deltocephalinae (Hemiptera, Auchenorrhyncha, Cicadellidae). Zootaxa, 2163: 1–64] the species should be placed within the genus Maiestas, and its proper name is Maiestas dorsalis (Motschulsky, 1859). Therefore, the wrong species name should be corrected throughout the text.
All other Latin names of species and genera should be accompanied by the author and date of descriptions when they are mentioned in the text for the first time.
Other detailed comments were put directly into the text.
